# Perioperative Management of Non-Small Cell Lung Cancer in the Era of Immunotherapy

**DOI:** 10.3390/cells14130971

**Published:** 2025-06-25

**Authors:** Ulas Kumbasar, Erkan Dikmen, Zeliha Gunnur Dikmen, Ates Tenekeci, Ilgen Mender, Sergei Gryaznov, Burak Bilgin, Saadettin Kilickap

**Affiliations:** 1Department of Thoracic Surgery, Faculty of Medicine, Hacettepe University, Ankara 06100, Türkiye; erkandikmen@hacettepe.edu.tr; 2Department of Medical Biochemistry, Faculty of Medicine, Hacettepe University, Ankara 06100, Türkiye; gunnur@hacettepe.edu.tr (Z.G.D.); atesktenekeci@gmail.com (A.T.); 3Department of Research and Development, MAIA Biotechnology, Inc., Chicago, IL 60606, USA; imender@maiabiotech.com (I.M.); sgryaznov@maiabiotech.com (S.G.); 4Medical Oncology, Yildirim Beyazit University, Ankara 06800, Türkiye; drbbilgin@hotmail.com; 5Medical Oncology, Liv Hospitals, Istinye University, Ankara 06800, Türkiye; skilickap@yahoo.com

**Keywords:** NSCLC, neoadjuvant therapy, adjuvant therapy, targeted therapy, immunotherapy

## Abstract

Non-small cell lung cancer (NSCLC) is the leading cause of cancer-related mortality worldwide. Nonetheless, deeper molecular understanding of NSCLC has resulted in novel therapeutic approaches, including targeted therapy and immunotherapy, which have improved patient prognosis and outcomes in recent years. Immune checkpoint inhibitors (ICIs), with or without chemotherapy, are now considered valuable components of treatment for NSCLC cases that do not have specific actionable genetic mutations. Patients with actionable genetic mutations are candidates for targeted therapies. The primary focus of this review is the rationale for using ICIs in the perioperative setting for patients with resectable NSCLC and in advanced disease settings. Furthermore, we compare the benefits of using ICIs with the challenges associated with their clinical implementation in resectable and advanced NSCLC. Finally, we emphasize the development of novel treatment strategies that potentially provide an optimal treatment choice for patients with advanced NSCLC.

## 1. Introduction

Lung cancer remains the main cause of cancer-related deaths worldwide, accounting for 18.4% of all cancer-related fatalities, resulting in a considerable social burden and economic loss. Non-small cell lung cancer (NSCLC) is the predominant subtype, comprising approximately 85% of lung cancer cases [1,2]. The other main subtype of lung cancer is small cell lung cancer (SCLC), which is less common than NSCLC.

NSCLC is an epithelial lung cancer, with the most common histological types being squamous cell carcinoma, large cell carcinoma, and adenocarcinoma [3]. Treatment strategies for NSCLC vary based on disease stage, molecular profile, tumor location, and other patient-specific considerations. Stages I and II are generally considered resectable NSCLC (rNSCLC), meaning the tumors can be surgically removed. While some Stage IIIA cases may also have resectable tumors, a significant portion of tumors—approximately 20% of newly diagnosed NSCLC cases—are unresectable due to distant metastatic disease and are therefore categorized as unresectable or advanced NSCLC [4]. Chemotherapy, radiation therapy, targeted therapy, immunotherapy, and combination strategies may be used in the treatment of both resectable and unresectable NSCLC, depending on the stage and molecular profile of the disease [3].

Over the past decade, immune checkpoint inhibitors (ICIs) and targeted therapies have significantly advanced treatment strategies for NSCLC, largely due to the availability of biomarkers, such as PDL1 levels within tumors for ICIs and EGFR mutations, ALK and ROS1 rearrangements, KRAS G12C mutations, and others for targeted therapies. These biomarkers help identify patients who may benefit from these treatments and ultimately improve patient outcomes across all stages. Monoclonal antibodies targeting the PD-1 (programmed cell death protein 1)/PD-L1 (programmed death-ligand 1) immune checkpoint axis, including atezolizumab, cemiplimab-rwlc, durvalumab, ipilimumab, nivolumab, pembrolizumab and tremelimumab, are used for the treatment of NSCLC. Local treatment strategies have also advanced for NSCLC, particularly in resectable disease, with improvements in surgical techniques [2,5,6].

These treatment modalities can be combined with surgery or administered as neoadjuvant or adjuvant therapy in patients with rNSCLC. In addition, in some cases, systemic therapy can be used in advanced diseases to reduce the tumor burden, potentially making surgery a viable option for these patients. Figure 1 shows a simplified workflow for clinical rNSCLC.

In this paper, we review the current state of the field for perioperative immunotherapy-before surgery (neoadjuvant), after surgery (adjuvant), or both-to improve outcomes in rNSCLC and available treatment options for advanced NSCLC. We also address the challenges associated with the clinical implementation of these therapies, the most recent advancements in clinical trials, and a comprehensive perspective on controversial issues. Finally, we highlight the development of an innovative treatment strategy that incorporates immune checkpoint inhibitors and telomere-targeting therapy, which has the potential to offer an optimal treatment option for patients with advanced NSCLC.

## 2. Potential Advantages of Treatment in the Neoadjuvant, Adjuvant, and Perioperative Approaches

In recent years, the neoadjuvant approach has been extensively studied in a variety of tumor types and is now widely employed in various malignancies because of its positive outcomes. Despite its limited use in the past for preparing inoperable patients for surgery, it has a broader range of applications today due to its many potential benefits. Preoperative approaches have been accepted as the standard of care for borderline operable patients as well as operable patients in various malignancies, most notably lung, breast, rectal, and gastric cancers.

One of the earliest recognized potential advantages of neoadjuvant therapy is its role in organ-sparing strategies. Particularly in diseases such as breast, larynx, and rectal cancers, neoadjuvant therapy can prevent deterioration in quality of life after organ loss and reduce the cosmetic impact by increasing the feasibility of organ-sparing or minimally invasive surgery. One of the most critical benefits of neoadjuvant treatment is the reduction in the frequency of pneumonectomy, which is one of the most important variables influencing life expectancy in patients with lung cancer [7]. In addition, the increased use of minimally invasive approaches may improve the patient’s quality of life in the postoperative period and reduce the risk of postoperative complications that may be associated with surgery.

Micrometastases have long been known to occur in various malignancies, and their presence has been associated with a poor prognosis, even in early-stage primary disease. Previous studies have shown that micrometastatic foci detected both in the lymph nodes and in the bone marrow have a negative effect on prognosis, especially in lung cancer [8,9]. Circulating tumor cell (CTC) analysis and similar techniques are now recognized as viable methods for the detection of micrometastases. Although discussed in detail in the following sections, the relationship between CTC positivity, recurrence, and poor prognosis is likely to be related to the presence of micrometastatic disease [10]. One of the possible benefits of neoadjuvant treatment is the potential for early systemic control after systemic treatment and the potential for clearance of micrometastatic foci in the preoperative phase.

Pathological complete response (pCR) can be defined as the lack of detection of malignant cells in the surgical specimen after neoadjuvant treatment. In particular, due to the favorable results of neoadjuvant studies of breast cancer, pCR was found to be an important marker that could predict survival, and thus, pCR has been recognized as an endpoint with accelerated approval for early-stage breast cancer by the Food and Drug Administration (FDA) in 2020 [11]. In addition to breast cancer, the pCR rate is an important parameter in clinical trials for several malignancies, particularly lung cancer. pCR is also significantly associated with survival, and its use as an alternative marker for survival has increased in recent years [12]. Consequently, achieving a pCR in lung cancer may be considered an important advantage of neoadjuvant therapy. The relationship between pCR and perioperative treatment in lung cancer will be discussed in more detail in the following sections.

Neoadjuvant ICI may induce a stronger immune response than adjuvant ICI, leading to improved clinical outcomes. Different mechanisms have been identified in in vitro and in vivo studies in different malignancies to support this hypothesis. One proposed explanation for the improved efficacy of neoadjuvant ICI is that the presence of the tumor in situ during treatment exposes T cells to a greater burden of tumor antigens. This may lead to more effective T cell activation and expansion, which is further amplified by immune checkpoint inhibition [13]. In a mouse model designed to evaluate the effects of neoadjuvant and adjuvant ICI-based therapies, mice receiving neoadjuvant anti-PD-1 in combination with anti-CTLA-4 showed improved survival compared to those treated with the same combination in the adjuvant setting. This improved survival was associated with increased infiltration of resected tumors by specific T cells, highlighting the potential of neoadjuvant ICI to induce a more robust tumor-specific immune response. Another explanation for the possible superiority of the neoadjuvant approach is postoperative immune changes. It is well known that surgery has a significant immunosuppressive effect on the patient’s immune system. Post-operative stress and inflammation often result in the release of inflammatory cytokines such as IL-6, IL-8, IL-10, and TNF, as well as the expansion of immunosuppressive cell subsets, including regulatory T cells (Tregs), tumor-associated macrophages (TAMs), and myeloid-derived suppressor cells (MDSCs) [14]. Due to the immunosuppressive environment in the postoperative setting, it is believed that initiating ICI treatment in the preoperative setting will lead to a better treatment response. The phase 2 SWOG 1801 trial in resectable stage III/IV melanoma demonstrated increased survival in neoadjuvant pembrolizumab compared to adjuvant pembrolizumab in the perioperative approach, despite the absence of a head-to-head trial in lung cancer [15]. These results support the concept that neoadjuvant treatment functionally inhibits the immune checkpoints on anti-tumor T cells prior to surgical removal. In another study supporting this hypothesis, a randomized phase 1b trial in 20 patients with stage III melanoma found that two cycles of neoadjuvant administration of nivolumab and ipilimumab prior to surgery resulted in greater expansion of tumor-resident T-cell clones compared to the same therapy given postoperatively [16]. Currently, neoadjuvant treatment strategies utilizing ICIs are gaining prominence in several cancer types, including triple-negative breast cancer, bladder cancer, and microsatellite instability-high (MSI-H) colorectal cancer [17,18].

Alongside the benefits of neoadjuvant treatment, certain potential drawbacks may arise during the treatment process. These potential disadvantages of the neoadjuvant approach include patient reluctance to undergo surgery based on their response to treatment (particularly in malignancies other than rectal cancer, where a non-surgical approach is accepted), disease progression during neoadjuvant treatment resulting in a missed window period for resection, and toxicity from neoadjuvant treatment that precludes the possibility of definitive surgery.

Adjuvant therapy is a treatment modality that has been used in oncology practice for many years. The rationale for adjuvant therapy is to eradicate macro- or micro-residual disease after surgery. The efficacy of adjuvant therapy has been extensively studied in almost all cancers, particularly lung cancer, and has become the standard of care after surgery for many cancers, depending on the stage of the disease. The LACE meta-analysis (Lung Adjuvant Cisplatin Evaluation) showed a 5.4% improvement in survival at 5 years for patients with NSCLC treated with cisplatin-based adjuvant chemotherapy. Recent studies have been published examining the efficacy of the ICIs such as atezolizumab, pembrolizumab, and durvalumab in the adjuvant setting; however, the outcomes of these trials are inconsistent [19,20,21].

Despite the continued acceptance of the adjuvant method in treatment guidelines, it is clear that the management of early-stage disease has transitioned to the neoadjuvant phase in recent years. However, there is increasing evidence supporting perioperative treatment approaches that include both the neoadjuvant and adjuvant phases. Numerous completed and ongoing trials of perioperative therapy strategies exist for breast cancer, gastric cancer, bladder cancer, and lung cancer. Perioperative treatment aims to take advantage of the benefits of neoadjuvant therapy, as previously outlined, while also providing the additional benefit of eradicating any residual disease with adjuvant therapy. There is growing debate over the necessity of adjuvant treatment for all patients, and a dynamic approach that incorporates factors such as neoadjuvant outcomes, tumor biology, and circulating DNA (ctDNA) analysis is gaining significance in perioperative management. The dynamic approach aims to improve survival by adding adjuvant treatment for selected patient populations while avoiding the medical, economic, and social burden of adjuvant treatment for certain patients.

## 3. Summary of Major Phase 2 and 3 Studies Investigating Neoadjuvant and Perioperative Treatments

Outcomes from multiple trials assessing neoadjuvant and perioperative ICI-based therapies demonstrated a substantial advantage of these strategies over neoadjuvant chemotherapy alone. However, the absence of data from any phase 3 trials that compare upfront surgery followed by adjuvant therapy with neoadjuvant or perioperative strategies has challenged clinicians to determine the optimal approach for patients with rNSCLC.

The primary objective of neoadjuvant therapy is to effectively eliminate micrometastatic disease at an early stage, thereby preparing patients for surgery and giving them a greater chance of it being curative. ICIs may exhibit greater efficacy in the neoadjuvant context, especially when localized tumors are intact and release an increased quantity of neoantigens, hence enhancing the immunological response and likely demonstrating minimal clonal resistance. Furthermore, undamaged lymph nodes are also anticipated to enhance the priming of immune cells [22].

The ICI era for rNSCLC commenced with Forde et al.’s single-arm pilot study [23]. In this trial, neoadjuvant nivolumab induced a major pathological response (MPR) in 45% of resected tumors with few adverse effects. Two phase 2 trials, NADIM and the research by Shu et al., subsequently validated these findings within the context of chemoimmunotherapy combinations. The NADIM trial demonstrated that neoadjuvant chemotherapy combined with nivolumab followed by surgery and adjuvant nivolumab significantly improved progression-free (77.1% (95% CI 59.9–87.7) at 24 months) and OS rates (69.3% (95% CI 53.7–80.6) at 5 years) in patients with resectable stage IIIA NSCLC with an acceptable safety profile. Shu and colleagues also demonstrated that neoadjuvant atezolizumab combined with chemotherapy achieved a high MPR rate (57%) and promising long-term survival (median disease-free survival of 34.5 months) with manageable toxicity in patients with rNSCLC [24,25].

Several phase 3 trials, including CheckMate 816, AEGEAN, Neotorch, KEYNOTE-671, CheckMate 77T, and RATIONALE 315, have reported results showing a consistent EFS (event-free survival) and OS (overall survival) benefit of combining ICIs with chemotherapy in the neoadjuvant setting [26,27,28,29]. Consequently, neoadjuvant chemoimmunotherapy is recommended as the preferred option instead of upfront surgery for medically operable resectable clinical stage IIIA or IIIB NSCLC. The evidence also indicates that chemoimmunotherapy may offer a significant therapeutic benefit for patients with clinical stage II cancer, albeit it is less advantageous than for those with stage III disease [6].

The integration of immune checkpoint inhibitors in the adjuvant context has also enhanced patient outcomes in recent years. Two phase 3 randomized trials, IMpower010 and PEARLS/KEYNOTE-091, compared adjuvant platinum-based doublet chemotherapy followed by ICIs with platinum-based doublet followed by optimal supportive care or placebo, respectively, and demonstrated enhanced disease-free survival (DFS) in the ICI cohorts [19,30]. However, subgroup analysis of the Impower 010 study revealed that this benefit did not extend to patients with PD-L1 levels less than 1%. The DFS advantage was predominantly observed in patients with PD-L1 levels of 50% or more. In contrast, the PD-L1 subgroup analyses of PEARLS/KEYNOTE-091 showed a significant benefit in DFS in the overall trial population (14% of which had PD-L1 levels < 1%). Nevertheless, the difference in DFS did not reach significance in patients with PD-L1 levels more than or equal to 50% [30].

KEYNOTE-671 is so far the sole trial in the neoadjuvant, perioperative, or adjuvant ICI scenario to demonstrate an OS benefit, with a favorable hazard ratio (HR) of 0.74 for stage IIIA and 0.69 for stage IIIB disease [26].

Adjuvant platinum-based chemotherapy is established as the standard of care for completely resected stages IB to IIIA NSCLC due to its demonstrated OS advantage. Thus, it is crucial to emphasize that for patients who require adjuvant immunotherapy, chemotherapy should be an integral component of the treatment regimen to maximize its efficacy [6].

When considering the data from the mentioned trials, it is important to keep in mind that there are several differences in the designs of these trials that can influence the interpretation of the studies, including the TNM stage classification edition, tumor PD-L1 expression percentage, and outcome definitions (DFS, EFS, OS, etc.). Moreover, while neoadjuvant and perioperative trials use clinical staging, adjuvant trials use pathologic staging, limiting direct comparison of the outcomes.

In conclusion, the majority of the existing trial data is still in its early stages, necessitating longer-term follow-up to determine the efficacy of these therapies regarding OS, which is the main objective for these patients in a curative context. A summary of the phase 3 trials in the perioperative and neoadjuvant settings is shown in Table 1.

## 4. Clinical Importance of Pathological Complete Response, Major Pathological Response, and ctDNA Clearance

### 4.1. Pathologic Response

pCR and major pathological response (MPR) are important parameters in neoadjuvant and perioperative studies. pCR is defined as the absence of viable tumor cells in surgically removed specimens, while MPR is defined as less than or equal to 10% viable tumor on pathological assessment. pCR and MPR have been used primarily in trials as prognostic markers, but there is increasing evidence that they can also be used as predictive markers for a dynamic treatment approach. In both neoadjuvant and perioperative studies, the development of pCR and MPR has been associated with a good prognosis. Table 2 presents the main findings of the studies that are pertinent to pCR and MPR.

In all of the above studies, the pCR and MPR rates were significantly better in the ICI-chemotherapy combination arm, and survival was significantly better in patients with pCR or MPR. However, several important questions remain unresolved. The key question is whether pCR or MPR should be considered a surrogate measure for survival. It is well known that the FDA has accepted pCR as a surrogate marker for early breast cancer trials, but no such approval currently exists for lung cancer trials. Currently, it is essential to accurately define the impact of pCR on survival, as the treatment strategy depends on the existence or absence of pCR. Numerous studies are currently underway to determine whether pCR serves as a surrogate marker in neoadjuvant or perioperative ICI-based therapies. A review of seven randomized controlled studies by Hines et al. indicated that pCR and MPR could be used as surrogate markers for two-year EFS; however, a similar assumption cannot yet be made for OS (the R^2^ of pCR and MPR with 2-year EFS was 0.82 (0.66–0.94) and 0.81 (0.63–0.93), respectively; for 2-year OS, the R^2^ for pCR and MPR were 0.55 (0.09–0.98) and 0.52 (0.10–0.96), respectively). Nonetheless, it was observed that assessing overall survival over an extended follow-up period may be more appropriate [36]. Similarly, in the systematic review and individual patient data meta-analysis conducted by Marinelli et al., both pCR and MPR were identified as predictive markers for EFS (pCR versus no pCR, HR: 0.13, 95% CI: 0.08–0.21; MPR versus no MPR, HR: 0.18, 95% CI: 0.13–0.26) [37]. Sugiyama et al. stated that pCR is a significant surrogate marker for patient-level surrogacy based on these results; nevertheless, there is no evidence to suggest that pCR serves as a trial-level surrogate measure. Therefore, they argued that EFS and OS should still be used as the main endpoints [12]. In conclusion, it is evident that patients with pCR and MPR experience improved survival outcomes. However, methodologically, further follow-up and additional analyses may be required to ascertain whether these markers serve as surrogate markers for survival.

In addition to pCR and MPR, pathological dynamic changes have been increasingly discussed in recent years. A pan-tumor scoring system, known as immune-related pathological response criteria, has been established to assess pathological response characteristics, including immune-mediated tumor regression. This quantitative system is applicable across different disease sites and assigns a residual tumor volume (RTV) score ranging from 0 to 100%. Deutsch et al. published the results of an exploratory analysis of the CheckMate 816 trial investigating the relationship between residual tumor volume and survival [31]. In this study, the patient group was categorized according to RTV percentage, and an inverse correlation was found between RTV percentage and EFS (patients with RVT-PT 0–5%, >5–30%, >30–80%, and >80% had 2-year EFS rates of 90%, 60%, 57%, and 39%, respectively). This study not only demonstrates the impact of depth of response on survival but also shows that patients who do not achieve a pCR cannot be categorized into a single survival group. The application of RVT as a substitute for pCR or MPR requires further research and will be explored in greater detail.

### 4.2. Circulating Tumor DNA Clearance

ctDNA refers to extracellular, non-encapsulated DNA fragments released by cancer cells into the bloodstream. Although the mechanism of ctDNA formation is not fully understood, it is thought to occur as a result of the spilling of cellular debris into the bloodstream due to apoptosis resulting from increased cell turnover. ctDNA clearance is defined as the complete disappearance or a significant reduction of detectable tumor-specific genetic material in plasma during or after treatment. It serves as a dynamic biomarker reflecting treatment response, particularly in the context of immunotherapy and targeted therapies, and holds promise for the early detection of minimal residual disease (MRD). The ctDNA clearance rate refers to the proportion of patients in whom circulating tumor DNA becomes undetectable in plasma following a defined course of therapy. ctDNA analysis is used for various purposes, ranging from mutation detection to guiding treatment decisions in oncology, and its applications continue to expand daily. Its use in early-stage cancers has been studied more intensively in recent years, and it has been shown to predict adjuvant treatment responses, particularly in colorectal cancer [38]. Recent years have seen a rise in evidence supporting the utilization of ctDNA in planning neoadjuvant or perioperative treatment strategies for early-stage lung cancer, with greater comprehension of its predictive and prognostic significance.

In the CheckMate 816 study, ctDNA levels were available in 25% of randomized patients, and 24% of all patients had detectable ctDNA levels at baseline. The ctDNA clearance rate was 56% in the nivolumab-chemotherapy arm and 35% in the chemotherapy arm. In the nivolumab-chemotherapy arm compared to the chemotherapy-only arm, a risk reduction of up to 70% for OS was observed in these patients with and without ctDNA clearance (HR: 0.31; (95% CI 0.10–0.90) [32]). In addition to survival, there was a dramatic correlation between ctDNA clearance and pathological response. In the nivolumab-chemotherapy arm, the pCR rate was 46% in patients with ctDNA clearance, while no pCR was achieved in patients without ctDNA clearance. In the perioperative CheckMate 77T study, a higher rate of ctDNA clearance was achieved in the perioperative nivolumab-chemotherapy arm (66% vs. 38%), and, similar to the CheckMate 816 study, the pCR rate was 50% in patients with ctDNA clearance, while no pCR was achieved in patients without ctDNA clearance. Despite the absence of statistically significant comparative analysis data, EFS was seen to be superior in patients exhibiting ctDNA clearance. In the nivolumab-chemotherapy arm, the 2-year EFS rate in patients with ctDNA clearance was 81%, while the 2-year EFS rate in patients without ctDNA clearance was 50% [34]. As shown in the perioperative analysis of the AEGEAN study, 65% ctDNA clearance was achieved after four cycles of neoadjuvant treatment in the durvalumab arm. In this study, ctDNA analysis was performed after each cycle of neoadjuvant treatment, and the importance of ctDNA clearance time was investigated. According to the results of the analysis, the ctDNA clearance rates after the 2nd, 3rd, and 4th cycles were 31%, 55%, and 65%, respectively. Analysis of the relationship between ctDNA clearance and pCR revealed that when ctDNA turned negative by the fourth cycle in patients with baseline ctDNA positivity, pCR was achieved in all patients, and MPR was observed in over 93% of patients. On the other hand, the absence of early ctDNA clearance was associated with a low rate of pCR (negative predictive value 89% for cycle 2 and 100% for cycle 4). Simultaneously, ctDNA clearance at early neoadjuvant timepoints was found to be associated with improved EFS [35].

Nevertheless, the clinical use of ctDNA remains subject to ongoing concerns and uncertainties. Its integration into routine clinical practice remains challenging. One major limitation is assay variability, as differences in sensitivity, specificity, and detection thresholds across platforms may lead to inconsistent results [39]. Furthermore, the timing of blood collection and the consistency of pre-analytical processes substantially influence ctDNA detectability, especially in the perioperative context where tumor burden is minimal. The accessibility of ctDNA testing poses a significant issue; high expenses and restricted availability in non-academic institutions prevent its wider adoption.

All these studies demonstrate that ctDNA is a strong predictive indicator of a treatment response. However, as indicated in the AEGEAN study, both the positivity or negativity of ctDNA and the timing of its clearance influence the disease’s progression. Therefore, there is still a long way to go in ctDNA analysis. Currently, the strategy of predicting treatment response based on ctDNA results is gaining popularity, yet it is obvious that additional research is required. A further challenge to address is the limited accessibility to ctDNA, particularly in socio-economically disadvantaged countries, such as those classified as developing countries. Consequently, efforts should be made to improve its technical and financial accessibility.

## 5. Biomarkers That May Affect Treatment Response and Potential Resistance Mutations

### 5.1. Programmed Death Ligand-1 (PD-L1)

PD-L1 is the most widely used biomarker, especially in lung cancer. Despite continuing uncertainties over PD-L1 usage, it is recognized as a predictive and prognostic biomarker, particularly in metastatic and locally advanced stages. Conflicting results have been obtained for PD-L1 in early-stage lung cancer, especially in the adjuvant phase [40]. In neoadjuvant and perioperative lung cancer studies, PD-L1 has been used as a stratification factor in treatment strategies, and the data obtained from the studies are generally consistent.

In the subgroup analysis of the CheckMate 816 trial, survival was significantly better in the patient group with PD-L1 levels ≥ 50%, while in other patient groups, survival was better in the chemoimmunotherapy arm than chemotherapy alone. In the EFS analysis, the HR (hazard ratio) was 0.85, 0.58, and 0.24 for PD-L1 < 1%, 1–49%, and ≥50%, respectively. This indicates that higher PDL1 expression is associated with prolonged EFS. In addition, pathological responses varied according to PD-L1 level; the pCR rate was 14.1%, 23.5%, and 40% for PD-L1 < 1%, 1–49%, and ≥50%, respectively. It is important to note that the PD-L1 IHC 28-8 pharmDx assay (Dako) kit was used for PD-L1 analysis in the CheckMate 816 study [28].

The KEYNOTE 671 study, which used the PD-L1 IHC 22C3 pharmDx assay kit for PD-L1 analysis, demonstrated similar PD-L1 TPS (Tumor Proportion Score) levels. For PD-L1 level < 1%, 1–49%, and ≥50%, the HR for EFS was 0.75, 0.52, and 0.48, respectively. The corresponding HR for OS was 0.91, 0.69, and 0.55, respectively. Specifically, in the PD-L1 < 1% cohort, the HR for OS was 0.91, prompting a discourse on the effectiveness of immunotherapies in this patient population [33].

In the CheckMate 77T study, an additional perioperative investigation, the PD-L1 IHC 28-8 pharmDx assay (Dako) kit was employed for PD-L1 analysis. The HR by PD-L1 subgroup was 0.73, 0.76, and 0.23 EFS for <1%, 1–49%, and ≥50% PDL1, respectively. The corresponding pathological CR rate by PD-L1 level was 8.6%, 22.6%, and 45.3%, respectively. Results of subgroup analyses for overall survival have not yet been made available [6,34,36,41,42,43,44,45].

The VENTANA PD-L1 [SP263] kit was used to determine PD-L1 levels in the AEGEAN study. This study’s survival outcomes based on PD-L1 levels demonstrated differences when compared to previous research. For PD-L1 < 1%, 1–49%, and ≥50% in tumor cells, the corresponding EFS HR value was 0.76, 0.70, and 0.60, respectively. The corresponding pCR rate for each PD-L1 group was 5.8%, 11.4%, and 22.9%, respectively. In contrast to other studies, there was no significant difference between the HRs for EFS in patients with PD-L1 levels < 1% and 1–49%, and interestingly, the benefit of the combination was more limited in the PD-L1 ≥ 50% group compared to other studies. Another key point is that the pCR rate in the PD-L1 < 1% arm was very close to that of the placebo arm (5.8% vs. 4.3%). These differences may be due to the PD-L1 kit used. While other studies used the DAKO kit, the AEGEAN study used the VENTANA PD-L1 [SP263] kit [35,42,46].

The diversity among PD-L1 assays (e.g., 22C3, 28-8, SP263) in the neoadjuvant therapy of NSCLC introduces ambiguity in biomarker-driven patient selection. The use of different clones in different trials may hinder clinical decision-making due to the absence of complete concordance. Consequently, employing the same assay confirmed as a companion diagnostic for a particular immunotherapy drug is crucial for precise patient classification. Current harmonization initiatives seek to resolve inter-assay inconsistencies.

The effect of PD-L1 levels on neoadjuvant or perioperative treatment in early-stage NSCLC has also been investigated in meta-analyses. In the meta-analysis conducted by Banna et al., the therapeutic benefit in the PD-L1 < 1% cohort was markedly inferior compared to the other cohorts. Patients with negative PD-L1 tumor status had a higher HR for 2-year EFS (HR, 0.75; 95% CI, 0.62–0.91) compared to those with low (HR, 0.61; 95% CI, 0.37–0.71) or high (HR, 0.40; 95% CI, 0.27–0.58) PD-L1 (*p*: 0.005 for test of group differences) [47]. In another network meta-analysis by He et al., a significant difference in EFS was found when perioperative immunotherapy was compared with neoadjuvant chemotherapy in the PD-L1 < 1% group (HR, 0.72, 95% CI: 0.53–0.97), whereas no difference was found between neoadjuvant immunotherapy and neoadjuvant chemotherapy (HR 0.85, 95% CI: 0.47–1.6) [48]. Another meta-analysis also revealed improved EFS with neoadjuvant chemoimmunotherapy in the group with PD-L1 expression < 1% (HR, 0.74; 95% CI, 0.62–0.89; *I*2 = 0%) [45].

Despite variances in methodological aspects and research outcomes throughout the aforementioned studies, a consistent conclusion is that the efficacy of ICI chemotherapy employing a neoadjuvant and perioperative approach is more limited in the PD-L1 < 1% group. The pCR rate in patients with PD-L1 < 1% was nearly equivalent to placebo in the AEGEAN trial, and the HR for OS was 0.91 in the PD-L1 < 1% group of the KEYNOTE 671 trial, which indicates a controversy on the efficacy of combination therapy for this patient population. It should be noted that the data obtained in these studies are based on subgroup analyses, and subgroups may not have sufficient power to be statistically evaluated. Furthermore, considering the numerically higher pCR and survival rates relative to placebo, the present findings do not allow us to conclude that ICI is ineffective in the PD-L1 < 1% group.

### 5.2. Other Biomarkers Besides PD-L1 Level

Studies on predictive or prognostic biomarkers, aside from PD-L1 levels, are currently in progress. Nevertheless, these results are predominantly derived from phase 2 trials.

In a post hoc analysis of the NADIM study, lower T-cell receptor clonality was associated with numerically lower OS. In addition, a Tumor-Immune Prognostic Score of three or higher was also significantly associated with improved progression-free survival (PFS) and OS. This study also investigated the effect of PD-L1 and tumor mutation burden (TMB) on survival, but neither PD-L1 levels (TPS ≥ 1% cut-off) nor TMB at diagnosis were found to be associated with PFS or OS [49].

In a different biomarker analysis from the NADIM study, patients with pCR exhibited overexpression of pathways associated with TCR (T cell receptor) co-expression, lymphocyte infiltration, type II interferon signaling, and antigen processing, and downregulation of pathways linked to tumor markers, proliferation, or PD-1 signaling. In addition, the absolute levels of follicular helper T cells, activated NK cells, M1 macrophages, resting dendritic cells, and immune cells were higher in pCR patients than in non-pCR patients [50].

The translational study by Molina-Alejandre et al. within the NADIM study cohort examined the influence of HLA class 1 antigen on treatment response, revealing no significant differences in pCR or survival between the HLA-deficient and HLA-proficient groups. However, patients with a high tumor mutation burden and HLA-proficient tumors had longer survival. Another finding of this study was that responders had more CD4+ T cells with HLA class II colonization and activated CD8+ T cells [51].

In the phase 2 TD-NeoFOUR study, perivascular CD4+ T cell infiltration, M1 macrophages, PD-1+ CD8+ T cell/CD8+ T cell ratio, and CD4+Foxp3+ Treg cells showed a significant change compared to pretreatment in the group that achieved a pCR after perioperative scintilimab and neoadjuvant antolinib chemotherapy [52]. Another study that examined pathological specimens before and after treatment in patients who underwent surgery after neoadjuvant treatment also reported that the CD3, FOXP3+, and CD8+/PD-1+ ratios may be predictors of a response to neoadjuvant ICI chemotherapy [53].

These studies show that there may be other biomarkers, beyond PD-L1, that can be used to predict a response to treatment in the neoadjuvant or perioperative setting. Particularly, analyses of the tumor microenvironment suggest that it may influence treatment response, as seen in the metastatic setting. However, further research is needed on biomarkers that may predict a response to ICI, particularly tumor mutation burden.

## 6. Potential Resistance Mutations

Several studies have shown that patients with metastatic NSCLC who have STK11, KEAP1, or both mutations and are treated with ICI tend to have reduced survival [54]. Consequently, STK11 and KEAP1 have been recognized as biomarkers indicative of immunotherapy resistance in individuals with metastatic NSCLC. Although these mutations have been studied extensively in advanced NSCLC, there are very few studies investigating their effect on treatment response in early lung cancer. In the post hoc analysis of the NADIM study, the presence of *STK11*, *KEAP1*, *RB1*, or *EGFR* mutations was found to be associated with worse PFS, but no statistically significant differences in overall survival were observed. Specifically, individuals with the KEAP1 mutation showed a markedly unfavorable clinical prognosis, with three out of four patients (75%) carrying this mutation experiencing disease progression and ultimately mortality [49]. The impact of these mutations on treatment responsiveness, particularly in the perioperative setting, warrants further investigation.

## 7. Treatment-Related Side Effects

Alongside efficacy, side effects constitute a significant endpoint in clinical trials. In diseases characterized by prolonged survival, such as early-stage lung cancer, it is crucial to preserve quality of life. In neoadjuvant and perioperative immunotherapy trials, the combination of ICIs plus chemotherapy is generally safe, well tolerated, and does not increase the safety burden.

In the CheckMate 816 study, the incidence of adverse events was comparable in both groups (92.6–97.2%), with grade 3–4 adverse events being somewhat higher in the chemotherapy group. The most common adverse events were hematological adverse events, particularly neutropenia. Treatment-emergent adverse events of any grade leading to discontinuation were similar in both arms (10.2% vs. 9.7%). Immune-related adverse events were mostly grade 1–2. Delayed surgery due to adverse events was slightly more common in the chemotherapy group than in the chemoimmunotherapy group (5.1% vs. 3.4%). The rate of surgery-related complications, which is another concern, was higher in the chemotherapy arm (46.7% vs. 41.6%) [28]. In the KEYNOTE 671 study, the rate of any grade adverse events was comparable in both groups; however, the incidence of grade 3 and higher adverse events was higher in the pembrolizumab group (17.7% vs. 14.3%, respectively). The most common grade 3 or higher adverse events were hematological. The rate of discontinuation due to treatment-emergent adverse events was higher in the pembrolizumab arm (12.6% vs. 5.3%) [20]. In the AEGEAN study, the incidence of treatment-related adverse events of any grade was slightly higher in the durvalumab arm (86.8% vs. 80.7%), while the incidence of grade 3 or higher was similar (42.4 vs. 43.2%). The rate of discontinuation due to treatment-related adverse events was higher in the durvalumab arm (12% vs. 6%). Immune-mediated adverse events of any grade were reported in 23.7% of patients receiving chemo-immunotherapy and 9.3% of patients receiving chemotherapy [35,42,46]. In the CheckMate 77T study, similar to other studies, the occurrence of any grade adverse events was comparable in both groups; however, the incidence of grade 3–4 adverse events was elevated in the nivolumab group (32.5% vs. 25.2%). Treatment-related adverse events leading to discontinuation were also more common in the nivolumab arm. The occurrence of adverse events during the neoadjuvant phase was comparable in both groups; however, in the adjuvant phase, adverse events were more common in the nivolumab group. The rate of surgery-related adverse events was similar in both arms. However, the incidence of adverse events resulting in delay of surgery was slightly higher in the nivolumab group. (3.5% vs. 2.2%) [6,34,36,41,42,43,44,45].

Meta-analyses and phase 3 trials demonstrated that the incorporation of immunotherapy has minimal influence on the incidence of side effects [55,56]. Nonetheless, some meta-analyses indicated that, while there was no significant difference in the frequency of all-grade adverse events and grade 3–4 adverse events, the occurrence of serious side effects and treatment discontinuation was higher in the perioperative immunotherapy group [57].

All these trials indicate a little rise in the occurrence of side effects when immunotherapy is incorporated into neoadjuvant or perioperative treatment, with this increase attributed to the perioperative phase of treatment. Upon examining the impact of side effects on surgical procedures, none of the trials identified any influence on the duration or feasibility of surgery.

## 8. Which Patient Should Continue Adjuvant Treatment?

The current controversial issue in neoadjuvant therapy strategy for early-stage NSCLC is whether adjuvant therapy is necessary for all patients. The primary factors driving this argument include cost, a notable rise in the incidence of side effects, particularly during the perioperative phase, and the perception that it may not enhance survival, especially among some selected patient populations.

Perioperative treatment is more expensive than the neoadjuvant approach. Cost has been recognized as a concern deserving equal consideration to medical toxicity. The cost disparity between the neoadjuvant technique and the perioperative strategy may be as much as fivefold. Consequently, the perspective that patient selection constitutes a cost-effective strategy for cost reduction has recently gained popularity. An additional significant component is the occurrence of adverse events. In the CheckMate 77T study, no difference in adverse events was observed across arms during the neoadjuvant phase; however, in the adjuvant phase, adverse events were considerably elevated in the ICI arm. Whereas not an exact comparison, the CheckMate 816 study exhibits a reduced rate of grade 3 and higher adverse events, especially pneumonitis, when compared with the perioperative studies [58].

Recent investigations have intensified to identify which patients require solely neoadjuvant treatment and which necessitate perioperative treatment. Two primary indicators for treatment selection that are currently being highlighted are pCR and ctDNA clearance. pCR seems to be the most emphasized potential marker. As previously stated, patients achieving a pCR exhibit markedly improved survival rates in both the chemotherapy and combined treatment groups. Specifically, in the CheckMate 816 trial, the 4-year OS rate in patients with pCR was 95%, and the 2-year EFS rate was 93% [32]. Regarding these superior survival outcomes—attainable only with neoadjuvant treatment in patients achieving a pCR—it has been widely recognized that perioperative treatment seems unnecessary for this population.

However, it is still controversial whether pCR is a surrogate marker for survival in early-stage NSCLC. The study conducted by Hines et al. examined whether pCR and MPR are surrogate markers for EFS and OS, and it was determined that pCR and MPR are surrogate markers for 2-year EFS (R^2^ of pCR and MPR with 2-year EFS were 0.82 [0.66–0.94] and 0.81 [0.63–0.93]), respectively; however, their significance for OS remains unestablished due to insufficient data maturity (R^2^ of pCR and MPR were 0.55 [0.09–0.98] and 0.52 [0.10–0.96], respectively) [36]. Furthermore, there is no clear evidence that pCR serves as a surrogate measure at the trial level. Consequently, time is required for the data to mature and for additional investigations to be undertaken. Nonetheless, pCR is well recognized as a prognostic factor.

The most concrete data comparing the perioperative and neoadjuvant treatment approaches is the landmark analysis comparing the CheckMate 77T and CheckMate 816 trials presented by Forde et al. at the 2024 World Lung Cancer Congress [43]. In this study, perioperative treatment provided up to a 40% risk reduction in EFS compared to neoadjuvant treatment (HR: 0.59, 95% CI 0.38–0.92). In the subgroup analysis, the EFS benefit of perioperative treatment was significantly better in patients without a pCR (HR 0.65, 95% CI 0.40–1.06). In patients with a pCR, the survival curves were very similar, and although there was a 42% risk reduction, the statistical significance was questionable because the confidence interval was too wide (HR 0.58, 95% CI 0.14–2.4). A further subgroup analysis was performed according to PD-L1 level. In patients with PD-L1 < 1%, the perioperative treatment was more effective than the neoadjuvant approach, whereas in the PD-L1 ≥ 1% group, no significant difference in EFS was found between the two groups (HR 0.51 (95% CI 0.28–0.93) and HR 0.86 (95% CI 0.44–1.70) for the PD-L1 < 1% and ≥1% groups, respectively). One possible explanation for the significant difference between perioperative treatment and the neoadjuvant approach in the PD-L1 < 1% group, but not in the PD-L1 ≥ 1% group, may be that the proportion of patients achieving a pCR in the PD-L1 ≥ 1% group is approximately twice as high as in the PD-L1 < 1% group (pCR rate: 28% vs. 14%). In contrast, the systematic review and individual patient data meta-analysis by Marinelli et al. showed conflicting results, finding that continuing treatment in the adjuvant setting regardless of pCR status did not contribute to additional EFS compared with neoadjuvant alone. The HR for EFS was 0.80 (95% CI: 0.28–2.29; *p*: 0.8) when only the neoadjuvant approach was compared with the perioperative approach in patients with a pCR, while the HR was 0.89 (95% CI: 0.67–1.18; *p*: 0.4) in patients without a pCR [37].

Consequently, despite the conflicting findings of Marinelli et al., it is suggested that the perioperative strategy is preferable to the neoadjuvant approach, particularly in patients lacking a pCR, and it is recommended to continue with treatment in the adjuvant setting for this group of patients. However, there is still serious disagreement about continuing treatment in the adjuvant setting in patients with a pCR. Proponents of continuing treatment in the adjuvant setting emphasize that this analysis is not powerful enough to make a statistical evaluation due to the small number of patients with a pCR, and they also state that this landmark analysis has methodological limitations, so they advocate adjuvant treatment regardless of pCR status [59]. Conversely, proponents of withholding adjuvant treatment in patients with a pCR argue that the survival curves for this group overlap in the research, and there is no evidence demonstrating that perioperative treatment is statistically superior. Specifically, they highlight that the CheckMate 816 update study revealed a 4-year OS rate of 95% and a 2-year EFS of 93% in patients with a pCR, which is very impressive [58]. In conclusion, definite decisions cannot be made with the data currently available. Nonetheless, a more precise decision may be achievable by using additional markers alongside the presence or absence of pCR.

ctDNA clearance has long been studied in many malignancies, particularly lung cancer, and is recognized as a prognostic and predictive marker. Studies in NSCLC have also shown that ctDNA clearance is a prognostic marker at different stages [60]. As outlined above, patients exhibiting ctDNA clearance in early-stage NSCLC demonstrate enhanced survival, and this is recognized as a prognostic indicator. Nonetheless, there remains inadequate data about whether ctDNA clearance can serve as a prognostic marker for treatment requirement in the adjuvant context. In the post hoc analysis of the AEGEAN study, EFS was analyzed according to both ctDNA clearance and pCR status, and the group with both pCR and ctDNA clearance had the best survival, while the group with ctDNA clearance but no pCR had inferior survival (HR: 0.39, 95% CI 0.11–1.41). These findings suggest that the combined use of these two parameters may have an impact on survival and may be useful in the future to predict the need for adjuvant treatment [61].

In conclusion, based on the current data, adjuvant treatment can be considered the standard of care for patients who have not achieved a pCR after neoadjuvant treatment. Moreover, although definite data is lacking, it is believed that patients without ctDNA clearance may derive benefits from adjuvant treatment due to poorer survival outcomes compared to those with ctDNA clearance. In patients achieving a pCR, while a definitive strategy for adjuvant treatment remains unavailable pending new evidence, it is recommended to incorporate other factors such as PD-L1 status, ctDNA clearance, and patient-specific clinicopathological characteristics into the decision-making process.

## 9. Controversial Issues in Perioperative Treatment

### 9.1. Approach to Persistent N2 Patient

The approach to a patient with persistent N2 disease following neoadjuvant chemoimmunotherapy is highly debatable within the thoracic oncology community. Two major questions emerge that require resolution. 1. Will we conduct radiological or invasive restaging of patients after induction with chemoimmunotherapy? 2. Will we proceed with surgery if there is radiologically suspicious or pathologically confirmed persisting N2 disease?

No phase 3 neoadjuvant trials required surgical mediastinal restaging, and data on the efficacy of this practice in enhancing patient outcomes are insufficient. While certain centers may decide to routinely perform invasive mediastinal restaging with re-mediastinoscopy, this procedure can be exceedingly difficult, frequently associated with a significant incidence of complications, and may also pose a risk of prolonging surgical delays. Meanwhile, there is no clear data to support this practice [6].

According to the available data obtained from perioperative immunotherapy trials, radiologic restaging, particularly with CT imaging, is not a reliable parameter. Metabolic re-evaluation with PET-CT after induction therapy is also considered an unreliable tool due to the phenomenon of nodal immune flaring (NIF), which is defined as radiologically abnormal nodes identified on restaging imaging following induction therapy with ICIs, which are free of cancerous cells and demonstrate de novo non-caseating granulomas upon pathological examination. NIF is linked to an inflamed nodal microenvironment and distinct fecal microbiome genera, without any pathological or radiological tumor responses or immunotherapy-related toxicity [62]. However, Zhang et al. conducted a multicenter study to analyze the diagnostic efficiency of PET-CT for ypN2 disease after neoadjuvant chemoimmunotherapy and identified PET-CT N2 positivity as an independent prognostic factor compared with N2 negativity and thus encouraged the use of PET-CT as a restaging tool after induction chemoimmunotherapy [63]. Overall, contrast-enhanced CT is sufficient for assessing disease progression prior to surgery, and in the absence of radiographic progression following neoadjuvant chemoimmunotherapy, invasive mediastinal restaging is not routinely recommended [6].

Currently there is no clear data on whether we should pursue surgery in the face of persistent N2 disease. Therefore, our decision to proceed with the operation should depend on the balance between the drawbacks (risk of early treatment morbidity and mortality) and benefits of surgery (probability to achieve effective local therapy). More practically, for patients who have few comorbidities and a high likelihood of achieving R0 resection, surgical resection can be considered, albeit with an uncertain marginal advantage. Recently, a consensus recommendation paper from the International Association for the Study of Lung Cancer (IASLC) commented on this controversial topic and recommended that in the absence of disease spread, patients who remain operable and resectable should proceed to surgery. A multidisciplinary tumor board should be constituted for patients presenting signs of cancer progression or when the feasibility of surgery is uncertain [6].

### 9.2. Surgery—Technical Difficulties After Immunotherapy

In the context of surgery following chemoimmunotherapy, surgeons are likely to operate on a greater proportion of patients with clinically node-positive disease or larger tumors necessitating extensive pulmonary resections.

Two primary concerns arise when considering surgery following immunotherapy: the potential impact of ICI-related toxicities (IRAE) on surgical care and the possible effects of ICI that could complicate the procedure, including perivascular or hilar fibrosis, as well as the feasibility of employing minimally invasive techniques versus an open approach. IRAE are toxicities that we are not used to seeing after chemotherapy, such as endocrinopathies (hyper- or hypothyroidism), colitis, hepatitis, pancreatitis, and some other less common immune-related events. Despite the infrequency of these events (<5%), they may delay surgery and lead to peri- or postoperative complications.

The results of a recent large multicenter trial (LCMC3) showed that surgery following neoadjuvant atezolizumab is safe and well tolerated while yielding a 20% MPR rate and encouraging OS [64]. Other trials that combined induction chemotherapy with ICI, such as CK 816, KN 671, and NEOSTAR, also demonstrated a low incidence of IRAE [20,28,46]. A recent systematic review and meta-analysis of 28 randomized controlled trials (RCT) showed that the addition of ICI was not significantly associated with increased treatment-related deaths (OR 1.76, 95% CI 0.95–3.25; *p* = 0.073). Nevertheless, the addition of ICI increased the incidence of grade 3–4 treatment-related adverse events (OR 2.73, 95% CI 1.98–3.76; *p* < 0.0001), adverse events leading to treatment discontinuation (3.67, 2.45–5.51; *p* < 0.0001), and treatment-related adverse events of any grade (2.60 [1.88–3.61], *p* < 0.0001) [65]. Thus, surgeons have to be alert to the possibility of IRAE due to their potential significant postoperative implications. Thus, surgeons have to be alert to the possibility of IRAE due to their potential significant postoperative implications. Takada et al. systematically analyzed the impact of neoadjuvant ICIs on surgery and perioperative complications in 18 studies. They reported a 30-day mortality rate of 0–5.4%, with 11 out of 16 trials reporting a mortality rate of 0%. Notably, they also concluded that preoperative ICIs are safe from a surgical perspective [66].

The interpretation of imaging following neoadjuvant therapy can be misleading and may lead to an excessively extensive resection due to the difficulties in accurately predicting the degree of pathological response post-neoadjuvant therapy based on radiographic imaging. Neither CT nor PET-CT can really distinguish the true response to the therapy. According to the LCMC3 trial, only 36% of the patients who had a radiologic partial response actually had MPR on final pathology. Also, patients who had stable disease radiologically achieved a 17% MPR [64]. Surgeons should recognize that a residual mass or lymphadenopathy after immunotherapy may indicate fibrosis rather than a viable tumor. In this setting, multiple frozen section analyses and intraoperative consultations with the pathologist might aid the surgeon in modifying the resection extent according to the residual tumor while minimizing the likelihood of a pneumonectomy for a patient with a pathological complete response wherever feasible.

The most technically demanding feature of surgical resection after induction immunotherapy is perihilar/perivascular fibrosis, which complicates dissection and may eventually result in intraoperative bleeding. Currently, the majority of neoadjuvant trials do not address this issue; therefore, future research should carefully evaluate the incidence and severity of perihilar fibrosis, as its presence may influence surgical strategy, the degree of lung resection, and operative morbidity and mortality. Within the last decade, the thoracic surgical community has progressively shifted from thoracotomy to minimally invasive techniques, including VATS and RATS, for the resection of early-stage NSCLC. As a result, a significant proportion of patients in multiple neoadjuvant trials received resections utilizing minimally invasive procedures. The conversion from minimally invasive techniques to open thoracotomy may indicate a surgically complex procedure. Conversion rates differ throughout research, with the maximum rate documented being 53.8%, indicating a significant level. Vascular incidents and difficulty dissecting the fibrotic hilum were deemed the primary causes for intraoperative conversion to thoracotomy [28,42,64,65,66,67]. As surgeons acquire greater proficiency in minimally invasive surgery, particularly in RATS, within these complex procedures, it will be crucial to evaluate whether these approaches will also correlate with enhanced outcomes after surgery.

Another intraoperative challenge that thoracic surgeons may encounter after neoadjuvant immunotherapy is hilar/mediastinal immune nodal flare, which has been reported in up to 16% of cases. Although it is primarily attributed to an inflammatory response following neoadjuvant immunotherapy, it can pose a challenge to the surgical procedure by mimicking metastatic spread and necessitating intraoperative pathological examination [62,68].

In summary, as neoadjuvant immunotherapy becomes a cornerstone of cancer treatment, the demand for highly skilled surgical expertise capable of performing challenging resections, often using minimally invasive techniques, increases. This expertise must be supported by robust multidisciplinary coordination to ensure optimal patient selection. While immunotherapy may complicate surgical procedures, the rate of perioperative complications and mortality rates are still deemed acceptable. A meticulously planned resection strategy is essential, and cautious surgical techniques in a collaborative fashion between surgeon and pathologist should be employed to prevent intraoperative events, conversion to open thoracotomy, and pneumonectomy. Further information regarding intraoperative findings from prospective controlled trials is necessary to develop guidelines and recommendations for thoracic surgeons.

## 10. Addressing Advanced NSCLC

Following the discussion on resectable NSCLC, attention must also be given to the management of advanced NSCLC, an area in which significant unmet medical needs remain. Current first-line treatment decisions for advanced NSCLC depend on several factors, including PD-L1 expression, and the presence of actionable mutations (e.g., EGFR, ALK, ROS1). However, most patients do not harbor these mutations and have treatment options limited to cytotoxic chemotherapy and/or immunotherapy. In untreated advanced NSCLC, first-line with PD-L1 expression ≥ 50% and no genetic mutation, pembrolizumab achieved significantly better OS compared to the chemotherapy group in the KEYNOTE-024 trial. The estimated percentage of patients alive at 6 months was 80.2% in the pembrolizumab group and 72.4% in the chemotherapy group [69]. However, despite the success of ICIs, most advanced NSCLC patients eventually develop resistance to ICIs and have very limited treatment options once they progress. In the multicenter, double-blind, randomized phase 3 trial REVEL, NSCLC patients who had progressed on a first-line platinum-based chemotherapy regimen received either docetaxel and placebo or docetaxel and ramucirumab (anti-VEGFR-2) combination in the second-line setting. This trial demonstrated a 64% disease control rate (DCR) and 23% overall response rate (ORR) compared to docetaxel (DCR 53% and ORR 14%). In addition, OS was 10.5 months in the ramucirumab-docetaxel combination arm and 9.1 months in the docetaxel plus placebo arm with an improvement of 1.4 months. PFS was 4.5 months in the ramucirumab-docetaxel combination arm and 3.0 months in the docetaxel plus placebo arm with an improvement of 1.5 months. However, 79% of patients treated in the combination arm developed grade ≥ 3 TEAE; therefore, there is an urgent need for safer and more effective treatment options [70]. In the third-line setting, currently, there is no standard of care for patients without genetic mutations, and therefore, various chemotherapy agents are being utilized, with reported DCRs in the 25–35% range [71].

## 11. ICI Reuse in Advanced NSCLC Patients Who Progressed on ICI

Despite the success of ICIs in NSCLC patients with stages I–IV, many patients do not respond to treatment, or those who initially respond may develop acquired resistance. New strategies have been explored to overcome ICI resistance, including the combination of ICIs with chemotherapy and radiation. Targeting different IC receptors—such as TIM-3 and TIGIT—in conjunction with anti-PD1 therapy also improved the overall immune response, potentially overcoming ICI resistance. The use of nivolumab for targeting PD-1 plus ipilimumab for targeting CTLA-4 as first-line therapy for unresectable NSCLC has been approved by the FDA. Pembrolizumab and ipilimumab as second-line therapy also showed successful outcomes in certain patients with stage IIIB/IV NSCLC [72].

Increased DNA damage by radiotherapy and chemotherapy increases the presence of cytoplasmic DNA and promotes neo-antigen production, which triggers anti-tumor immunity. New agents that target DNA damage response (DDR) pathways combined with ICIs have been investigated in clinical trials. These trials include cyclin-dependent kinase 4/6 inhibitors (CDK4/6i), poly (ADP-ribose) polymerase inhibitors (PARPi), ataxia telangiectasia inhibitors, Rad3-related (ATR) kinase inhibitors, WEE1 inhibitors, checkpoint kinase 1 (CHK1) inhibitors, and DNA-dependent protein kinase (DNA-PK) inhibitors [73]. Another promising strategy involved combining ICIs with telomere-targeting therapy, a treatment that targets telomerase-positive cancer cells.

## 12. Phase 2 Study of Telomere-Targeting Agent THIO Sequenced with Cemiplimab in ICI-Resistant Advanced NSCLC

Telomerase is an attractive target because over 80% of all cancers and approximately 78–83% of all NSCLC types are telomerase positive, whereas most normal somatic cells do not express telomerase [74,75]. Ateganosine (THIO; 6-thio-2′-deoxyguanosine) is a small molecule, first-in-class direct cancer telomere targeting agent that selectively kills telomerase-positive cancer cells. THIO is incorporated into de novo synthesized telomeres, leading to chromatin uncapping, DNA damage signal generation, and rapid apoptosis [76]. Preclinical data in NSCLC indicates that low doses of THIO induce sensitivity to ICIs when administered prior to an ICI in tumors that otherwise are resistant to ICI. Sequential treatment with THIO and ICI showed a potent and durable anti-tumor activity in preclinical models [77]. THIO kills cancer cells via dual mechanisms of action: (a) induction of DNA damage pathway results in the formation of Telomere Dysfunction Induced Foci (TIF), followed by rapid G2/M arrest or cell death of telomere-positive cancer cells; and (b) activation of cGAS-STING-dependent innate and adaptive immune responses upon release of damaged tumor DNA resulting from THIO treatment.

THIO-101 is a Phase 2, randomized, dose-optimization clinical study in adults with advanced NSCLC without genetic mutations who either progressed or relapsed after one to four prior treatment lines including ICI alone or in combination with platinum chemotherapy (NCT05208944). Using a modified 3 + 3 design, the safety lead-in (Part A) enrolled 10 patients who received ateganosine (THIO) 360 mg IV (120 mg daily on days 1, 2 and 3), followed by 350 mg cemiplimab on day 5, administered every three weeks. Following completion of Part A, enrollment was opened in the dose-finding portion of the study (Part B). Using a Simon 2-stage design, 79 patients were assigned to 360, 180, or 60 mg of THIO followed by cemiplimab every three weeks for up to 1 year in Part B. Patient disease status was assessed at cycle 3 day 1, cycle 5 day 1 and subsequently every 9–12 weeks. The trial completed enrollment in Parts A and B in February 2024, and the ateganosine dose of 180 mg/cycle was proven to be the most efficacious and selected as the optimal dose for further clinical development. The trial expansion is scheduled to begin in 2025 for third-line patients, comprising two arms: ateganosine as a single agent and ateganosine sequenced with libtayo (Figure 2).

### 12.1. Safety Findings from Parts A and B of the THIO-101 Study

Based on the data presented at the SITC meeting in 2024 (data cut-off: 16 September 2024), and ELCC meeting in 2025 (data cut-off: 15 January 2025), 79 patients were enrolled and received at least one dose of THIO [78,79]. THIO plus cemiplimab was reported to be well tolerated in this heavily pretreated NSCLC population, with most events ranging from grade 1 to 2 in severity. Most treatment-emergent adverse events (TEAE) were mild or moderate in severity. The majority of TEAE events observed laboratory value elevations, except nausea (12.7% overall and 2.4% at the 180 mg dose) and decreased appetite (3.8% overall and 2.4% at the 180 mg dose). No DLTs (dose limiting toxicities) have been reported in Part A during the safety lead in. A related grade ≥ 3 ALT (alanine aminotransferase) increase was reported in nine patients (11.4%), including two patients receiving 360 mg, four patients receiving 180 mg, and three patients receiving 60 mg. No clinical symptoms were associated with the elevated ALT laboratory values, and all ALT levels returned to baseline or normal with no lasting effects. All other related grade ≥ 3 events occurred in <5% of patients. Following an event of grade 4 liver function tests (LFT) elevation in a patient receiving 360 mg in Part B, enrollment in the 360 mg arm was paused. Enrollment was completed in Part B at the selected dose of 180 mg/cycle in February 2024.

### 12.2. Efficacy Findings from Parts A and B of the THIO-101 Study

Partial responses (PRs), as defined by RECIST 1.1, were reported for nine subjects (six patients in the second-line group, three patients in the third-line group), with seven PRs confirmed by a second scan per the investigators’ assessment (four patients in the second-line group, three patients in the third-line group) [78,79]. Thirty-three patients achieved survival follow-up beyond 12 months (20 patients in the second-line group, 19 patients ongoing follow-up; 13 patients in the third-line group, 7 patients ongoing follow-up and 1 patient has received 29 cycles of therapy, ongoing) [79]. Among these, 14/22 (63%) patients crossed the 5.8-month OS threshold [80] and 17/22 (77%) patients crossed the 2.5-month PFS threshold [81,82]. In the third-line setting with THIO (n = 22), the estimated median overall survival (OS) was at 16.9 months with a 95% CI (confidence interval) lower bound of 12.5 months and a 99% CI lower bound of 10.8 months, suggesting sustained benefit from the treatment in this advanced, heavily treated population [79]. In the third-line setting, DCR was 77% in 22 patients who received any dose of THIO, which is higher than the reported DCRs in the 25–35% range for chemotherapy [71,79].

### 12.3. Biomarker Findings from Parts A and B of the THIO-101 Study

This trial identified circulating tumor cells (CTCs) to analyze telomere dysfunction-induced foci (TIF) in CTCs. TIF-positive CTCs were characterized as the fraction of cells expressing both TRF1 (telomeric protein) and gammaH2AX (DNA damage marker) in both CD326 (epithelial cell adhesion molecule, EpCAM) and PanKRT (pan-cytokeratin)-expressing CTCs. TIF analysis demonstrated the intended on-target mechanism of action by showing the modification of telomeres in CTCs by THIO. On average, patients in the stable disease (SD) and partial response (PR) groups showed increased levels of biomarker TIF, whereas the progressive disease (PD) group did not demonstrate a statistically significant increase in the TIF biomarker. This indicates that TIF formation in CTCs was shown to be a good biomarker of on-target activity [78]. Additionally, interleukin-6 (IL-6) was also evaluated to assess the immune response to THIO in NSCLC patients receiving third line and beyond treatment. IL-6 was elevated on cycle 1 day 5 after THIO treatment in patients responding to THIO and cemiplimab treatment. This suggests that the initial increase in IL-6 may be associated with an immune response to THIO and cemiplimab treatment, suggesting its potential as a biomarker to predict treatment response [79]. CTCs were also marked by PDL1 to evaluate PDL1 status on cycle 1, day 1 (C1D1, baseline) in NSCLC patients receiving sequential treatment of THIO and cemiplimab as third-line and beyond treatment. The response to THIO and cemiplimab, demonstrated by partial response and stable disease, was independent of baseline PDL1 status [79].

THIO sequenced with cemiplimab has durable activity in this hard-to-treat patient population (ICI-resistant and chemotherapy-resistant progressors). Induction of TIFs in CTCs from patients treated with THIO shows an on-target effect, suggesting a potential link between biomarker TIF positivity and more favorable clinical outcomes. An initial elevation of IL-6 may be associated with the immune response to THIO and cemiplimab, indicating its potential as a predictive biomarker for treatment efficacy. Based on the clinical trial, the 180 mg dose was selected as the best dose of THIO, as it showed better safety and superior efficacy compared with other doses. The safety profile of THIO has the potential to be significantly better than chemotherapy, providing a potential advantage to give treatment longer, which usually translates into longer survival. As there is no established clinical third-line treatment for advanced NSCLC without driver mutations, for patients who progressed or discontinued due to toxicity or relapsed after ICI therapy, THIO has demonstrated promising results in addressing unmet medical needs so far.

## 13. Conclusions

Numerous investigations are still ongoing to investigate the new treatment strategies in early-stage and advanced NSCLC. ICI-based therapies have been evaluated in neoadjuvant, adjuvant and perioperative (neoadjuvant and adjuvant) settings in early-stage NSCLC. These studies primarily attempt to enhance pCR and survival rates through novel combinations, as well as to assess the efficacy of treatment intensification, particularly in patients who have not achieved a pCR following neoadjuvant therapy. Even though neoadjuvant therapy has safety and efficacy comparable to or better than adjuvant therapy, adjuvant therapy has its own benefits. Therefore, the basis of decision making for neoadjuvant, adjuvant or perioperative therapies depends upon the stage of disease and the availability of biomarkers. The therapeutic options for advanced NSCLC are limited to systemic therapy when surgery is not a viable choice. The identification of driver mutations has significantly impacted NSCLC treatment; however, only a small subset of patients carries these mutations. Immunotherapies have made a substantial impact on NSCLC, offering broader therapeutic benefits. However, the occurrence of intrinsic or acquired resistance has led to the development of various combination strategies. THIO is one of the leading examples of these strategies, where it is combined with ICI and has shown promising results in a population with an unmet medical need. While other combination strategies have been developed or are currently under investigation, identifying the right treatment for the right patients—those most likely to benefit—remains essential in NSCLC treatment.

## Figures and Tables

**Figure 1 cells-14-00971-f001:**
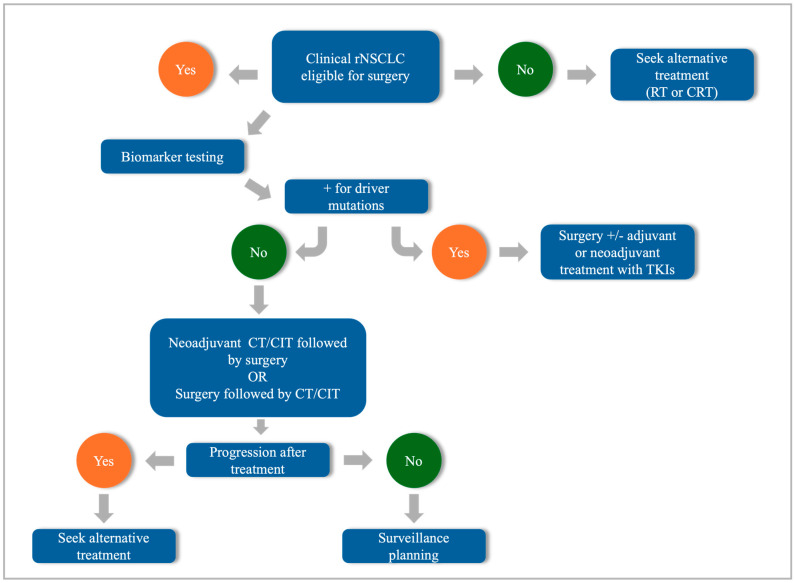
Workflow for rNSCLC. rNSCLC: resectable non-small cell lung cancer; RT: radiotherapy; CRT: chemoradiotherapy; TKI: tyrosine kinase inhibitor; CIT: chemoimmunotherapy.

**Figure 2 cells-14-00971-f002:**
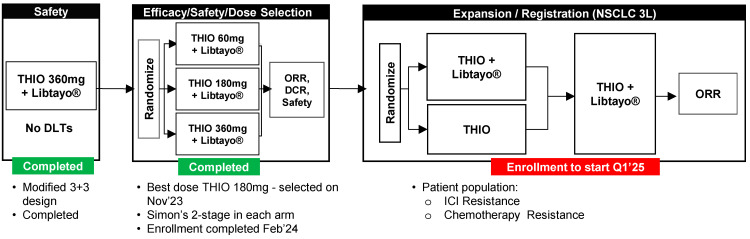
Schematic of the THIO-101 clinical trial design (79). Primary endpoints: safety, ORR (objective response rate), and DCR (CR, PR, and SD). Secondary endpoints: DoR (duration of response), PFS (progression-free survival), and OS (overall survival). Exploratory endpoints: PK and PD (activity of THIO in CTCs (circulating tumor cells) measured by specific biomarkers). The schematic has been adapted from the ELCC poster, 2025.

**Table 1 cells-14-00971-t001:** Summary of phase 3 trials in the perioperative and neoadjuvant settings.

Study	N	Cycle	Stage	Primary End Points	Adj	FU (Months)	pCR (%) CT-ICI vs. CT	EFS (Months)	PD-L1 Assay	EFS HR (95% CI, *p*)	OS HR (95% CI, *p*)
CM-816	358	3	IB–IIIA	EFS, pCR	−	29.5	24 vs. 2.2	31.6–20.8	IHC 28-8	0.66 (0.49–0.9)	0.71 (0.47–1.07, *p* = 0.045)
KN-671	786	4	II–IIIB	EFS, OS	+	36.6	18.1 vs. 4	47.2–18.1	IHC 22C3	0.59 (0.48–0.72)	0.72 (0.56–0.93, *p* < 0.01)
AEGEAN	802	4	II–IIIB	EFS, pCR	+	11.7	17.2 vs. 4.3	NR–25.9	IHC SP263	0.68 (0.53–0.88, *p* < 0.01)	NR
NeoTORCH	404	3 + 1	II–IIIB	EFS, MPR	+	18.2	24.8 vs. 1	NR–15.5	IHC JS311	0.40 (0.28–0.57, *p* < 0.01)	0.62 (0.38–0.99, *p* = 0.05)
CM-77T	461	4	IIA–IIIB	EFS	+	25.4	25.2 vs. 4.7	NR–18.4	IHC 28-8	0.58 (0.42–0.81, *p* < 0.00025)	NR

CM: CheckMate; KN: KEYNOTE; Adj: adjuvant; FU: follow-up; pCR: pathological complete response; EFS: event-free survival; OS: overall survival; HR: hazard ratio; 95% CI: 95% confidence interval; NR: not reported; CT: chemotherapy; ICI: immune checkpoint inhibitor (CheckMate 816 and CheckMate 77T: nivolumab-carboplatin/cisplatin; KEYNOTE 671: pembrolizumab-cisplatin; AEGEAN: durvalumab-carboplatin/cisplatin; NEOTORCH: toripalimab-carboplatin/cisplatin).

**Table 2 cells-14-00971-t002:** Main findings of the studies that are pertinent to pCR and MPR.

Study	Treatment	pCR Rate	MPR Rate	Key Findings
CheckMate 816 [31] (preop, preliminary results)	Nivo + CT	24%	36%	Patients with MPR and pCR had better EFS than those without.
CheckMate 816 [32] (preop, 4-year update)	Nivo + CT	-	-	OS significantly increased in patients with pCR (HR: 0.08 [0.02–0.34]; 4-year OS rates, 95% vs. 63%).
KEYNOTE 671 [33] (periop)	Pembro	18.1%	30.2%	Patients who achieved MPR and pCR had better EFS than those who did not.
CheckMate 77T [34] (periop)	Nivo	25.3%	35.4%	Survival was significantly better in patients with pCR in both the nivo and CT arms. HR for pCR vs. no pCR: 0.20 (95% CI, 0.08–0.50) in the nivo group; 0.41 (95% CI, 0.13–1.30) in the CT group.
AEGEAN [35] (periop)	Durva	17.2%	33.3%	Durva was effective in terms of pCR across all subgroups. EFS was better in patients with pCR.
Hines et al. [36] (review of 7 RCTs)		-	-	pCR and MPR could be used as surrogate markers for two-year EFS (R^2^ of pCR and MPR with 2-year EFS was 0.82 (0.66–0.94) and 0.81 (0.63–0.93), respectively). Similar assumption not yet for OS. Assessing OS over extended follow-up may be more appropriate.
Marinelli et al. [37] (SR and meta-analysis)		-	-	Both pCR and MPR identified as predictive markers for EFS (pCR vs. no pCR, HR: 0.13, 95% CI: 0.08–0.21; MPR vs. no MPR, HR: 0.18, 95% CI: 0.13–0.26).
Deutsch et al. [31] (exploratory analysis of CM 816)		-	-	Inverse correlation between RTV percentage and EFS. Patients with RVT-PT 0–5%, >5–30%, >30–80%, and >80% had 2-year EFS rates of 90%, 60%, 57%, and 39%, respectively.

preop: preoperative; periop: perioperative; pCR: pathologic complete response; MPR: major pathological response; EFS: event free survival; OS: overall survival; nivo: nivolumab, pembro: pembrolizumab; durva: durvalumab; CT: chemotherapy; RCT: randomized controlled trial; SR: systematic review; HR: hazard ratio; CM: CheckMate; RTV: residual tumor volume.

## Data Availability

No new data were created or analyzed in this study.

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
