# Peer review of "Perioperative Management of Non-Small Cell Lung Cancer in the Era of Immunotherapy"

_cells, 2025, doi:10.3390/cells14130971_

Round 1
Reviewer 1 Report
Comments and Suggestions for Authors
I congratulate authors for this review article.
The article presented is well written and explained.
Only a few minor issues are present:
1: Improve abstract.
2: Check keywords.
3: Try to improve the introduction.
4: Section 2 needs more recent studies.
5: Sections 4.1 and 4.2 lack basic references for some critical statements.
6: Improve section 9.2.
7: A conclusion section need to be added.
Author Response
I congratulate authors for this review article. The article presented is well written and explained.
Thank you for the constructive feedback. We hope that the revised version reflects a significant improvement.
Only a few minor issues are present:
1: Improve abstract.
We improved the abstract.
2: Check keywords.
We updated the keywords.
3: Try to improve the introduction.
We have included additional details in the introduction, and we believe these enhancements and edits have improved the introduction.
4: Section 2 needs more recent studies.
Thank you for your suggestions. The section you mentioned has been revised with references to current studies.
5: Sections 4.1 and 4.2 lack basic references for some critical statements.
Thank you for your suggestions. In the section you mentioned, all major studies related to pCR and ctDNA in the neoadjuvant and perioperative settings have been discussed, and appropriate references to these studies have been provided.
6: Improve section 9.2.
Section 9.2. has been improved according to your comment.
7: A conclusion section need to be added.
A conclusion section has been added to the manuscript.
Reviewer 2 Report
Comments and Suggestions for Authors
Dear Authors
You have written about an interesting issue in lung cancer treatment, that is the neoadjuvant immunochemotherapy versus neoadjuvant chemotherapy. You prove your point of better results with immunochemotherapy.
However, the article is not well written and it is difficult to read, because there is excessive text on issues that should go into a Table. For example 4.1 Pathologic response can be replaced by a table.
Furthermore Table 1 is not clear.
I really desliked your concept of financial toxicity. I would suggest to remove it, because it does not get along with the rest of the paper.
You have a tendency to mention issues that you do not explain. For example:
"A pan-tumor scoring system, known as immune-related pathological
response criteria, has been established to assess pathological response characteristics,
including immune-mediated tumor regression." I think you should explain what this is about.
You do not explain what ctDNA clearance rate is. You should.
You come up with the NADIM study as if every reader should know what it is about.
Conclusions: this is an interesting paper poorly written.

Author Response
Dear Authors
You have written about an interesting issue in lung cancer treatment, that is the neoadjuvant immunochemotherapy versus neoadjuvant chemotherapy. You prove your point of better results with immunochemotherapy.
However, the article is not well written and it is difficult to read, because there is excessive text on issues that should go into a Table. For example 4.1 Pathologic response can be replaced by a table.
Furthermore Table 1 is not clear.
Thank you for your valuable and proper comments. A table showing the relevant studies has been added to the section 4.1. and table 1. has been revised to be more clear for the readers.
I really desliked your concept of financial toxicity. I would suggest to remove it, because it does not get along with the rest of the paper.
Thank you for your suggestions. In line with your recommendations, the section related to financial toxicity has been removed from the manuscript.
You have a tendency to mention issues that you do not explain. For example:
"A pan-tumor scoring system, known as immune-related pathological
response criteria, has been established to assess pathological response characteristics,
including immune-mediated tumor regression." I think you should explain what this is about.
Thank you for your comment. However, the definition and application of the scoring system are provided in the following sentences, and the exploratory analysis results from the CheckMate 816 trial, which utilized this scoring system, have also been included. Therefore, at this stage, we do not consider further elaboration necessary within the scope of our manuscript
You do not explain what ctDNA clearance rate is. You should.
Thank you for your suggestions. As you indicated, an explanatory section on ctDNA clearance and ctDNA clearance rate has been added.
You come up with the NADIM study as if every reader should know what it is about.
An informative explanation of the NADIM study has been added to the relevant section.
Conclusions: this is an interesting paper poorly written.
Reviewer 3 Report
Comments and Suggestions for Authors
Dear Authors,
his review presents a comprehensive and timely synthesis of recent developments in the perioperative management of non-small cell lung cancer (NSCLC) in the context of immunotherapy. The manuscript is well-structured, covering the rationale, clinical evidence, biomarkers, resistance mechanisms, and future directions in both resectable and advanced NSCLC. It is a valuable contribution to the literature and offers clarity in an evolving field. However, there are a few areas that require clarification or improvement to enhance the clarity and scientific rigor in this manuscript. My comments are described as follows:
Comments:
- The authors successfully outline the objectives of the review. However, a clearer distinction between discussions of resectable versus unresectable/advanced NSCLC would enhance readability.
- Consider summarizing the structure of the review at the end of the introduction for better navigability.
- The manuscript provides an excellent overview of major trials (e.g., CheckMate 816, KEYNOTE-671, AEGEAN), including data on pCR, EFS, and OS. However, a summary table with direct comparison of trial design elements (e.g., PD-L1 assay used, staging, endpoints) would be valuable to readers for quick reference.
- The ctDNA clearance section is informative. However, it would benefit from a critical appraisal of the limitations in using ctDNA for treatment stratification (e.g., assay variability, accessibility).
- The PD-L1 and alternative biomarker sections are well-written. The discussion on the variability of results across different PD-L1 assays (e.g., 28-8 vs. SP263) is important but could be better integrated with recommendations on clinical utility.
- The review of surgical implications post-ICI is comprehensive. However, more emphasis could be placed on the need for surgical expertise and multidisciplinary coordination, especially as minimally invasive techniques continue to evolve.
- Consider a schematic diagram summarizing treatment pathways (e.g., neoadjuvant, perioperative, adjuvant) with decision nodes based on biomarkers.
- Please cite the references in the manuscript under color booking.

Author Response
Dear Authors,
This review presents a comprehensive and timely synthesis of recent developments in the perioperative management of non-small cell lung cancer (NSCLC) in the context of immunotherapy. The manuscript is well-structured, covering the rationale, clinical evidence, biomarkers, resistance mechanisms, and future directions in both resectable and advanced NSCLC. It is a valuable contribution to the literature and offers clarity in an evolving field. However, there are a few areas that require clarification or improvement to enhance the clarity and scientific rigor in this manuscript. My comments are described as follows:
Comments:
- The authors successfully outline the objectives of the review. However, a clearer distinction between discussions of resectable versus unresectable/advanced NSCLC would enhance readability.
Thank you for the positive feedback. We now included some information about the differences between resectable vs unresectable/advanced NSCLC.
- Consider summarizing the structure of the review at the end of the introduction for better navigability.
We expanded the introduction with more detailed information and summarized the structure of the review at the end of the introduction. We believe these revisions have strengthened the overall quality of the MS.
- The manuscript provides an excellent overview of major trials (e.g., CheckMate 816, KEYNOTE-671, AEGEAN), including data on pCR, EFS, and OS. However, a summary table with direct comparison of trial design elements (e.g., PD-L1 assay used, staging, endpoints) would be valuable to readers for quick reference.
Thank you very much for your valuable suggestion. In line with your recommendations, we have expanded and updated the table. We have also included a comparative discussion of the methodologies and outcomes.
- The ctDNA clearance section is informative. However, it would benefit from a critical appraisal of the limitations in using ctDNA for treatment stratification (e.g., assay variability, accessibility).
Thank you for your suggestions. In line with your recommendations, we have added a concise paragraph that provides a critical appraisal of ctDNA.
- The PD-L1 and alternative biomarker sections are well-written. The discussion on the variability of results across different PD-L1 assays (e.g., 28-8 vs. SP263) is important but could be better integrated with recommendations on clinical utility.
Thanks for your valuable suggestions. As you pointed out, an additional section has been included to highlight the use of different PD-L1 assays and to discuss the potential implications of this variability.
- The review of surgical implications post-ICI is comprehensive. However, more emphasis could be placed on the need for surgical expertise and multidisciplinary coordination, especially as minimally invasive techniques continue to evolve.
The relevant section has been revised according to your valuable comments.
- Consider a schematic diagram summarizing treatment pathways (e.g., neoadjuvant, perioperative, adjuvant) with decision nodes based on biomarkers.
A workflow diagram has been added to the manuscript accordingly.
- Please cite the references in the manuscript under color booking.
The relevant section has been cited.
Round 2
Reviewer 2 Report
Comments and Suggestions for Authors
No further comments
Reviewer 3 Report
Comments and Suggestions for Authors
Dear Authors,
Thank you for your revision. I have no more questions.